# Wnt/β-Catenin Signalling and Its Cofactor BCL9L Have an Oncogenic Effect in Bladder Cancer Cells

**DOI:** 10.3390/ijms23105319

**Published:** 2022-05-10

**Authors:** Roland Kotolloshi, Mieczyslaw Gajda, Marc-Oliver Grimm, Daniel Steinbach

**Affiliations:** 1Department of Urology, Jena University Hospital, 07747 Jena, Germany; roland.kotolloshi@med.uni-jena.de (R.K.); marc-oliver.grimm@med.uni-jena.de (M.-O.G.); 2Section of Pathology, Department of Forensic Medicine, Jena University Hospital, 07747 Jena, Germany; mieczyslaw.gajda@med.uni-jena.de

**Keywords:** bladder cancer, UTR mutation, *BCL9L*, Wnt Wnt/β-catenin signalling, cancer progression

## Abstract

Bladder cancer (BC) is characterised by a high recurrence and progression rate. However, the molecular mechanisms of BC progression remain poorly understood. BCL9L, a coactivator of β-catenin was mutated in the 5′ and 3′ untranslated regions (UTRs). We assessed the influence of UTRs mutations on *BCL9L*, and the role of *BCL9L* and Wnt/β-catenin signalling in BC cells. UTR mutations were analysed by a luciferase reporter. BCL9L protein was assessed by immunohistochemistry in BC tissues. Cell proliferation was examined by crystal violet staining and by the spheroid model. Moreover, migration and invasion were analysed in real-time using the xCelligence RTCA system. The A > T mutation at 3′ UTR of *BCL9L* reduces the luciferase reporter mRNA expression and activity. BCL9L is predominantly increased in dysplastic urothelial cells and muscle-invasive BC. Knockdown of *BCL9L* and inhibition of Wnt/β-catenin signalling significantly repress the proliferation, migration and invasion of Cal29 and T24. In addition, *BCL9L* knockdown reduces mRNA level of Wnt/β-catenin target genes in Cal29 but not in T24 cells. *BCL9L* and Wnt/β-catenin signalling play an oncogenic role in bladder cancer cells and seems to be associated with BC progression. Nevertheless, the involvement of BCL9L in Wnt/β-catenin signalling is cell-line specific.

## 1. Introduction

Bladder cancer (BC) was the tenth most commonly cancer worldwide in 2020, however, the ranking was raised to the seventh place in the male population. The incidence in men is higher than in women [1,2]. The main risk factor for BC is cigarette smoking, which accounts for 50% of all cases. Moreover, the occupational and environmental exposure to carcinogens such as aromatic amines, especially in industrial areas where workers are in contact with chemicals during the processing of various products like plastics, rubbers, paints, and other by-products, is the second most important risk factor [3,4,5]. BC originates from the urothelial layer of the bladder, and it is distinguishable as non-muscle-invasive bladder cancer (NMIBC) and muscle-invasive bladder cancer (MIBC), which correspond to 75 and 25%, respectively. Approximately 47% of patients with NMIBC experience recurrence and around 10–15% of them progress to MIBC. The probability of progression for patients at five years is up to 45% [6,7,8]. Progressive BC needs radical therapy and usually leads to metastasis and it is associated with poor prognosis and overall survival [9,10]. However, the molecular mechanisms of BC progression are not intensively investigated and it is crucial to identify progression-associated markers and targets for therapy against progressive BC.

The Wnt signalling pathway is involved in several cellular processes such as proliferation, differentiation, migration, polarisation and self-renewal during development. Wnt pathway is distinguished in the canonical Wnt pathway (referred as Wnt/β-catenin pathway) and non-canonical Wnt pathway, which is independent of β-catenin [11,12]. Wnt/β-catenin signalling, as one of the main signalling pathways, is disrupted in many diseases and cancer [13,14]. A limited number of reports have shown that Wnt/β-catenin signalling is correlated to BC progression, stage, invasiveness, and poor prognosis. Moreover, it is suggested that Wnt/β-catenin signalling is involved in the maintenance and regulation of urothelial cancer stem cells [15,16,17,18]. Normally, in the absence of Wnt ligands, the Wnt/β-catenin signalling is inactive. The cytoplasmic β-catenin protein is bound to the destruction complex, phosphorylated, and degraded. In the presence of Wnt ligands, the Wnt/β-catenin signalling is activated, in which β-catenin is translocated to the nucleus, which interacts with TCF/LEF transcription factor and other cofactors such as PYGO and B cell CLL/lymphoma 9/L (BCL9/L) coactivators to form a functional complex resulting in the activation of the signalling [19,20,21,22,23,24].

BCL9L protein contains three conserved regions known as homology domains: HD1, HD2, and HD3 [25]. The conserved HD2 domain of BCL9L protein is required to interact with the first armadillo repeat of β-catenin in the nucleus, while the HD1 domain of BCL9L binds to the homeodomain (PHD) of PYGO protein, a transcriptional activator of the Wnt/β-catenin signalling. Therefore, it is suggested that the BCL9L protein acts as an adaptor protein to link β-catenin with PYGO to activate β-catenin/TCF/LEF1-mediated transcription. However, BCL9L protein can also function through β-catenin independent of PYGO protein through another conserved transactivation domain in the C terminal region of the protein in some specific cell types [26,27,28]. In controversial to this, BCL9L protein has functions that are independent of the canonical Wnt/β-catenin signalling [29,30]. In cancer, *BCL9L*/BCL9L is overexpressed in several tumour entities and it is associated with the development of tumourigenicity, tumour stage, and cancer progression [28,31,32]. For example, *BCL9L* expression is increased in hepatocellular carcinoma (HCC). Moreover, knockdown of *BCL9L* negatively affects the malignant behaviour of HCC cells through reduction of Wnt/β-catenin signalling [31,33,34]. The mechanism of action of BCL9L is mostly based on the enhancement of the canonical Wnt/β-catenin signalling in order to further contribute to tumour progression [28,35,36,37,38,39,40]. However, in oestrogen receptor alpha positive breast cancer cell lines MCF7 and T47D, BCL9L promotes tumourigenicity independent of Wnt/β-catenin signalling, but through induction of the oestrogen receptor alpha signalling [32].

*BCL9L* gene has not yet been analysed in bladder cancer. *BCL9L* was identified to be mutated at the 5′ and 3′ untranslated regions (UTR) of BC patients in our previous study [41]. Therefore, we aimed to analyse whether the identified mutation could influence the expression of *BCL9L* as well as the functional role of *BCL9L* in BC cell lines. Given the known interaction between BCL9L and β-catenin proteins, we also aimed to decipher the role of BCL9L on the Wnt/β-catenin signalling in BC cells. The current study now suggests that BCL9L and ß-catenin also play an oncogenic role in bladder cancer.

## 2. Results

### 2.1. The Tumour-Associated Gene BCL9L Is Frequent Mutated in Bladder Cancer

From a previous study, the whole-exome sequencing analysis of progressive NMIBC and their corresponding MIBC of eight patients identified frequent mutations in the coding sequence as well as in the 5′ and 3′ UTRs of several genes to be part of distinct pathways, including Wnt/β-catenin signalling pathway [41]. The tumour-associated gene *BCL9L*, a coactivator of β-catenin, carries frequent mutations in the regulatory 5′ and 3′ UTR regions of mRNA transcript in three patients out of eleven. Furthermore, the Cancer Genome Atlas Study of Bladder Cancer 2017, whole-exome sequencing of 412 MIBC detected that *BCL9L* is mutated in 11 patients out of 412, indicating that *BCL9L* might have a role in bladder cancer [42].

It has been well described that single point nucleotide alterations or mutations at the 5′ and 3′ UTR can influence gene expression, suggesting an aberrant expression of tumour-associated genes in cancer [43,44,45,46,47]. The role of the UTR mutations of *BCL9L* gene in the regulation of gene expression was addressed by the luciferase reporter system. The whole wildtype or mutated 5′ UTR sequences (965 bp) was cloned upstream, while a fragment of wildtype or mutated 3′ UTR (783 bp) was cloned downstream of the luciferase reporter gene in the pmirNanoGLO vector (Appendix A). It was not possible to generate and clone the whole 3′ UTR sequence of *BCL9L* due to the big sequence size of 4540 bp. The double mutated 5′ UTR (G > T and A > T) does not influence the luciferase activity and luciferase mRNA in T24 cells (Appendix A). For TCCsup cells, although the double mutated 5′ UTR slightly reduced the luciferase activity (1.15-fold induction, *p* < 0.05), it did not affect the luciferase mRNA level. Interestingly, the single mutated 3′ UTR (A > T) of *BCL9L* significantly reduced the luciferase activity (16-fold reduction for T24 and 27-fold reduction for TCCsup) and luciferase mRNA expression (1.7-fold and 3.6-fold reduction in T24 and TCCSsup, respectively) compared to the wildtype sequence (*p* < 0.05, Appendix A). These results suggest that especially the A > T mutation at 3′ UTR could be responsible for an aberrant expression of *BCL9L* in bladder cancer.

### 2.2. BCL9L Expression Is Associated with Poor Survival and Bladder Cancer Progression

*BCL9L* is upregulated in hepatocellular carcinoma (HCC) compared to adjacent non-tumour areas. Moreover, a high level of *BCL9L* is correlated with poor overall survival of HCC patients, suggesting the role of *BCL9L* in the progression and poor prognosis of HCC [31]. Since *BCL9L* has not yet been linked with bladder cancer, the human protein atlas platform was utilised to investigate the association of *BCL9L* in bladder cancer. A high mRNA level of *BCL9L* is associated with lower survival of bladder cancer patients (*p* = 0.0029, Figure 1). Moreover, the 5-year survival for patients with higher expression of *BCL9L* was 9% and with *BCL9L* lower expression was 47%, respectively [42,48]. This suggests that overexpression of *BCL9L* is associated with poor survival and a high level of *BCL9L* might not be beneficial for patients with bladder cancer.

We analysed the mRNA expression of *BCL9L* in the matched microdissected NMIBC and MIBC tumour tissues of eight patients with progressive disease (patient no. 1 to 8, Table 1) as well as in three ureters from non-bladder cancer patients. The mRNA expression of *BCL9L* was significantly upregulated in MIBC compared to ureters from non-bladder cancer patients (2.9-fold induction, *p* < 0.05, Figure 2A). Furthermore, *BCL9L* mRNA expression was slightly increased in MIBC compared to NMIBC, however, the results are not statistically significant (Figure 2A).

The protein expression of BCL9L was analysed in the formalin-fixed paraffin embedded (FFPE) tissues from 11 MIBC and 7 matched NMIBC samples (Figure 2B–D, Appendix A). Overall, BCL9L was expressed very heterogeneously in tumour samples as well as in dysplastic and non-dysplastic urothelium. The BCL9L staining was observed both in the nucleus and in the cytoplasm but strongly intensified in the nucleus, especially in the case of high expression. In non-dysplastic urothelium, the expression is increased towards the superficial cell layers. The same is found in papillary tumour formations, especially the peripheral cell layers that express BCL9L, while inner tumour layers are often weak stained or negative. In MIBC, negative solid tumour formations sometimes are closed to positive tumour formations. The H-score was used for quantification of the immunohistochemical staining of BCL9L protein (Appendix A) [49]. The non-dysplastic urothelium showed a negative to moderate staining with an average H-score of 0.77 (patient 2, 7, 8, Appendix A). In dysplastic urothelium of cancer samples, BCL9L was weak to strongly expressed (mean H-score 1.08, Appendix A), apparently stronger expressed, as in non-dysplastic urothelium, especially in the higher grade dysplastic cells (patient 2, 8, Appendix A). The papillary primary non-muscle-invasive tumours expressed BCL9L very heterogeneously, predominantly in the peripheral cell layers with weak to moderate intensity, however, the average H-score was 0.63 (Appendix A). The distribution of the BCL9L staining in MIBC was homogeneous in nine patients. However, MIBC patients 3, 5, and 8 also contain negative tumour cell areas for BCL9L, which were in the neighbourhood of moderately stained areas. The average H-score of all MIBC was 1.12. Comparison of NMIBC and matched MIBC showed an increased expression of BCL9L in the MIBC in five of the seven cases (Figure 2C,D, Appendix A). Tumour cells with very strong expression of BCL9L were found in MIBC of patients 1, 5, 6, 9 (Appendix A). The data shows that expression of BCL9L protein was predominantly increased in dysplastic urothelium and MIBC compared to NMIBC and non-dysplastic urothelium, suggesting an association with aggressiveness and tumour stage of bladder cancer cells.

### 2.3. Knockdown of BCL9L Represses Proliferation Independent of Apoptosis in Bladder Cancer Cells

Given the association of *BCL9L* expression with BC progression, we investigated the functional role of *BCL9L* knockdown in BC cells. It remains unknown to our knowledge so far whether *BCL9L* has an oncogenic effect on BC cells. This was addressed by functional analysis of *BCL9L* in bladder cancer cells after knockdown experiments using siRNA technology. Cal29 and T24 cells were selected for analysis as a model for invasive bladder cancer cells. The mRNA level of *BCL9L* was significantly reduced in Cal29 (3-fold and 1.5-fold on day 2 and 3, respectively) and T24 (1.8-fold and 1.6-fold on day 2 and 3, respectively) after knockdown compared to siControl (*p* < 0.05, Figure 3A). The protein level was strongly reduced in both cell lines after 3 days of transfection of siBCL9L compared to siControl (Figure 3B), confirming knockdown by siBCL9L.

The effect of *BCL9L* on the proliferation of BC cells T24 and Cal29 was analysed by crystal violet staining. *BCL9L* knockdown significantly represses the proliferation of both Cal29 (2.1-fold reduction) and T24 (1.7-fold reduction) cells compared to siControl (*p* < 0.05, Figure 3C), suggesting that *BCL9L* could promote the proliferation of bladder cancer cells.

The reduction of cell proliferation after *BCL9L* knockdown could be through induction of cellular apoptosis. Here, the effect of the knockdown of *BCL9L* on apoptosis is investigated by V-FITC and PI staining and measures by flow cytometry in BC cells. Camptothecin treatment was used as a positive control. The early and late apoptotic or dead cells were significantly increased in camptothecin treatment compared to DMSO control in Cal29 and T24 cells (*p* < 0.05, Appendix A), confirming that the annexin V-FITC and PI staining kit can be utilised for detecting apoptosis in BC cells. Next, the effect of siBCL9L on the induction of apoptosis in T24 and Cal29 cells was analysed. The percentage of early apoptotic cells was similar in siBCL9L and siControl treatments, at about 3.8%, while the late apoptotic or dead cells were slightly, however, significantly reduced to 4.5% by siBCL9L compared to 7.4% siControl (*p* < 0.05, Figure 3D) in Cal29 cells, suggesting that siBCL9L might slightly reduce late apoptosis or cell death in Cal29. In the case of T24 cells, knockdown of *BCL9L* did not significantly influence apoptosis compared to control (Figure 3D). In conclusion, *BCL9L* knockdown by siRNA transfection repressed cell proliferation in BC cell Cal29 and T24 without induction of apoptosis.

### 2.4. Knockdown of BCL9L Reduces Migration and Invasion in Bladder Cancer Cells

We investigated the effect of *BCL9L* on cell migration and invasion in bladder cancer cells in real-time using the xCELLigence RTCA system after transfection with siBCL9L. *BCL9L* knockdown significantly inhibits the cell migration of Cal29 (*p*-value < 0.05 for the time frame 12–20 h) and T24 (*p*-value < 0.05 for the time frame 13–42 h) compared to siControl (Figure 4A,B, Appendix A). Knockdown of *BCL9L* significantly represses the cell invasion of Cal29 (*p*-value < 0.05 for the time frame between 5–21 h) and T24 (*p*-value < 0.05 for the time frame 34–42 h) cells compared to siControl (Figure 4C,D, Appendix A). Our data show that *BCL9L* knockdown inhibits migration and invasion of BC cells. It is suggested that *BCL9L* could promote cell proliferation, migration and invasion in BC cells.

### 2.5. The Influence of BCL9L on Wnt/β-Catenin Signalling Is Cell Line Specific in Bladder Cancer

It has been reported that BCL9L is localised in the nucleus of cells and interacts with β-catenin or PYGO proteins, enhancing Wnt/β-catenin signalling [28,50]. However, in the *mouse* system, bcl9l is localised in the cytoplasm, suggesting a cytoplasmic role of bcl9l in *mouse* cells [29]. The localisation of BCL9L in bladder cancer cells is not known. Therefore, the localisation of BCL9L was examined in Cal29 T24 and TCCsup cells by immunofluorescence. BCL9L is mainly localised in the nucleus of Cal29 cells, however, T24 and TCCsup cells show more cytoplasmic localisation (Figure 5), suggesting a distinct role of BCL9L in these cell lines.

Given the already known involvement of BCL9L in the Wnt/β-catenin signalling, mRNA expression of some Wnt/β-catenin target genes such as the Wnt/β-catenin regulator *AXIN2*, the transcription factors *LEF1* and *SP5*, the apoptosis inhibitor *BIRC5* and the metalloproteinases *MMP9* and *MMP14* were analysed after cotreatments of Wnt/β-catenin activator (SKL2001) and siBCL9L in Cal29 and T24 cells. Thereby the activator SKL2001 binds to AXIN2 protein and it interferes with the interaction of AXIN2 and GSK3-β proteins, resulting in the release of β-catenin from the destructive complex and consequently in the activation of Wnt/β-catenin signalling [51]. In Cal29 cells, the mRNA expression of *AXIN2* and *LEF1* target genes was significantly increased with the increase of the SKL2001 concentration, which was significantly reversed by the knockdown of *BCL9L*. *SP5* mRNA level was significantly increased after SKL2001 but siBCL9L was not able to reduce *SP5* expression. No differences were observed for the *BIRC5* and *MMP14* targets after SKL2001 treatment, however, co-treatment with siBCL9L led to a significant reduction of mRNA expression of *BIRC5*. *MMP9* mRNA was significantly increased after SKL2001 treatment and surprisingly was further induced after transfection with siBCL9L in Cal29 cells. Without ß-Catenin activation by SKL2001, the knockdown by siBCL9L only affected the expression of *AXIN2* and *MMP9* on a significant level (*p* < 0.05, Figure 6A).

In T24 cells, the mRNA level of *AXIN2*, *LEF1*, *MMP9*, and *MMP14* was significantly increased after SKL2001 treatment. No differences are observed in the case of the *SP5* target gene. Unexpectedly, *BIRC5* mRNA expression was significantly reduced with increased SKL2001 concentration. However, a positive effect of SKL2001 on target expression was not reversed after knockdown of *BCL9L.* Similar to Cal29 cells, *MMP9* mRNA expression was strongly and significantly increased in a concentration manner of SKL2001, and *BCL9L* knockdown resulted in further induction of *MMP9* in T24 cells (*p* < 0.05, Figure 6B). The data suggests that the activator SKL2001 could increase a subset of Wnt/β-catenin target genes at a higher level in Cal29 compared to T24 cells. Moreover, the SKL2001-induced expression of a subset target is reduced after the knockdown of *BCL9L* in CAL29.

In conclusion, the reduction of a subset of Wnt/β-catenin target genes in Cal29 but not in T24 cells after *BCL9L* knockdown supports that *BCL9L* could influence Wnt/β-catenin signalling only in Cal29 cells.

### 2.6. Inhibition of Wnt/β-Catenin Signalling Reduces Proliferation and Spheroid Growth in BC Cells

In line with the observation that *BCL9L* could influence Wnt/β-catenin signalling in Cal29 cells, we investigated the effect of Wnt/β-catenin signalling inhibition by the specific inhibitor iCRT3 on BC cells Cal29 and T24. iCRT3 binds to β-catenin protein and it interferes with the interaction between β-catenin and TCF transcription factors resulting in the inhibition of Wnt/β-catenin signalling [52]. To confirm the inhibition of Wnt/β-catenin signalling by the inhibitor iCRT3, the mRNA expression of Wnt/β-catenin target genes *AXIN2*, *LEF1*, *SP5*, *BIRC5*, *MMP9* and *MMP14* were analysed after treatment of 40 μΜ and 20 μΜ iCRT3 in Cal29 and T24 cells. In Cal29, the mRNA expression of all these targets is significantly reduced after iCRT3 treatment with stronger repression of *AXIN2*, *MMP9* (*p* < 0.05, Appendix A). In T24 cells, the mRNA level of *AXIN2*, *SP5*, and *BIRC5* is significantly reduced with a stronger effect to be observed at 40 μΜ iCRT3, however not so stronger as in Cal29 cells (*p* < 0.05, Appendix A). *MMP14* expression was significantly increased after iCRT3 treatment. In general, the effect of iCRT3 was stronger in Cal29 compared to T24 cells, suggesting a distinct response in bladder cancer cells. Nevertheless, the specific inhibitor iCRT3, especially at 40 μΜ concentration, could reduce a subset of Wnt/β-catenin target genes in Cal29 and T24, confirming that it can inhibit the Wnt/β-catenin signalling.

To examine the effect of the inhibition of the Wnt/β-catenin signalling by iCRT3 treatment on the proliferation of bladder cancer cells in a monolayer model, the T24 and Cal29 cells were treated with 40 μM and 20 μM iCRT3. iCRT3 significantly inhibits the proliferation of both Cal29 and T24 cells after 8 days treatment in a concentration-dependent manner compared to DMSO control (*p* < 0.05, Figure 7A). The influence of iCRT3 was also analysed in the 3D spheroid growth of T24 cells because spheroids mimic much better the natural three dimensional growth of tumour cells. Inhibition of Wnt/β-catenin signalling by the inhibitor iCRT3 significantly reduced the spheroid volume of T24 cells at different time points (*p* < 0.05, Figure 7B). Unfortunately, we were not able to analyse the effect of iCRT3 on Cal29 spheroids because Cal29 doesn’t form stable spheroids. Overall, the inhibition of the Wnt/β-catenin signalling by iCRT3 significantly suppressed the proliferation and spheroid growth of BC cells, suggesting that Wnt/β-catenin signalling could promote BC growth.

Furthermore, we investigated the effect of iCRT3 on the induction of apoptosis in BC cells Cal29 and T24 at 40 μM and 20 μM concentrations. 40 μM iCRT3 significantly reduces early apoptosis and late apoptosis/cell death in Cal29 compared to DMSO control (*p* < 0.05, Appendix A). In the case of T24 cells, treatments with 40 μM and 20 μM iCRT3 did not significantly influence apoptosis compared to control (Appendix A).

### 2.7. The Inhibition of Wnt/β-Catenin Signalling Suppresses Migration and Invasion in BC Cells

Next, we examined the effect of the specific inhibitor iCRT3 on the cell migration and invasion of BC cells Cal29 and T24 in real-time. For the cell migration and invasion, 40 μM iCRT3 was chosen because it showed the strongest effect on cell proliferation and spheroid growth. The 40 μM iCRT3 treatment significantly inhibits the cell migration of Cal29 (*p*-value < 0.05 for time frame 1–24 h) and T24 (*p*-value < 0.05 for time frame 9–14 and 39–42 h) compared to control (Figure 8A,B). Moreover, 40 μM iCRT3 treatment significantly suppressed the cell invasion of Cal29 (*p*-value < 0.05 for time frame between 11–24 h) and T24 (*p*-value < 0.05 for time frame between 15–42 h) cells (Figure 8C,D). These data show that inhibition of Wnt/β-catenin signalling by iCRT3 could suppress the cell migration and invasion of BC cells Cal29 and T24.

## 3. Discussion

Several studies from whole-exome DNA and RNA sequencing analysis have shown that bladder cancer has a high mutation load and have identified distinct pathways and genes to be mutated. However, it is essential to identify progression-associated genes and proteins as possible biomarkers or targets for the therapy against progressive BC [53,54,55]. In previous work, we identified frequent mutations in the coding sequence as well as in the 5′ and 3′ of UTR of several tumour-associated genes to be part of distinct pathways and protein families [41]. The tumour-associated gene *BCL9L*, as a coactivator of β-catenin is involved in Wnt/β-catenin signalling, was identified to be exclusively mutated at the 5′ and 3’ UTR of BC patients. Here, we showed that one mutation (A > T) at 3′ UTR of *BCL9L* could influence the luciferase reporter activity in BC cells, suggesting that this mutation might be responsible for an aberrant expression of *BCL9L*. Other studies have shown that mutations at the 5′ and 3′ UTR of several genes might have an impact on gene expression and are strongly associated with various tumour entities [43,44,45,46,47]. Thus, the UTR mutations could provide a molecular mechanism for an aberrant expression of tumour-associated genes such as *BCL9L*.

It has been well demonstrated by some studies that *BCL9L* is overexpressed in several cancer types and is associated with tumour progression. BCL9L protein is overexpressed and correlated with poor prognosis and poor overall survival in various tumour entities, suggesting the role of *BCL9L* in the cancer’s progression [31,32,38,56,57]. In bladder cancer, *BCL9L* has not yet been analysed and it is not known whether it plays a role in BC progression. Survival analysis of BC patients indicates that high mRNA level of *BCL9L* is significantly correlated with worse overall survival in bladder cancer patients, suggesting that a *BCL9L* high level might not be beneficial in BC patients. Furthermore, we showed that *BCL9L* mRNA expression was significantly increased in MIBC compared to normal ureters. Interestingly, BCL9L protein was increased on average as well as in some subpopulations of cells in MIBC and in dysplastic urothelial cells compared to NMIBC and non-dysplastic urothelium, suggesting an association with BC tumour stage and malignancy of carcinoma cells. In addition, the localisation of BCL9L was observed to be both in the nucleus and cytoplasm of tumour cells but with stronger staining in the nucleus, in agreement with its role as transcriptional cofactor of Wnt/β-catenin. Furthermore, we observed that BCL9L protein is mainly localised in the nucleus of Cal29 cells compared to T24 and TCCsup, where it is stronger expressed in the cytoplasm. This agrees with the observation that BCL9L is expressed in the nuclei of invasive lesions in breast cancer tissues, suggesting that strong staining of BCL9L in the nucleus might be related to cancer progression [58]. We hypothesise that the increased protein expression of BCL9L in dysplastic urothelial cells and especially in MIBC, could be responsible for the invasiveness and progression of BC. However, a limitation of the current work is the low number of patients that was analysed, and an increase of tumour sample cohort is strongly required.

Next, we focused on addressing whether *BCL9L* has any functional role in BC cells in vitro. Knockdown of *BCL9L* significantly repressed the proliferation, migration, and invasion in T24 and Cal29 cells, without induction of apoptosis, maybe through a cell-cycle arrest mechanism. Here, we suggest that *BCL9L* has an oncogenic role in BC cells. Epithelial-mesenchymal transition (EMT) plays a crucial role in the migration and invasion of tumour cells. Therefore, the influence of BCL9L on EMT has to be analysed in further studies using 3D cell culture or organ models. In general, the oncogenic effect of *BCL9L* in cancer is also confirmed by other studies for several tumour entities. In breast cancer, depletion of *BCL9L* by knockdown experiments inhibits the cell proliferation of MCF7 cells, while overexpression of *BCL9L* promotes tumourigenicity in transgenic *mice*, confirming the oncogenic effect of *BCL9L* in cell culture and animal models [32]. Other in vitro and in vivo studies have proved the oncogenic effect of *BCL9L* through knockdown and overexpression experiments in other cancer entities [31,38,50,56,59]. Overall, we propose that *BCL9L* could promote the malignant behaviour of BC.

Moreover, we focused on investigating the potential molecular mechanism of how *BCL9L* could promote malignancy in BC. It is known that BCL9L protein interacts with β-catenin and activates the Wnt/β-catenin signalling, which is involved in tumourigenesis [28,35]. After activation of the Wnt/β-catenin pathway, β-catenin interacts with TCF/LEF transcription factor and other factors such as BCL9L to activate the signalling inside in the nucleus [14,60,61]. Dysregulation of β-catenin cofactors including the BCL9L protein, alter Wnt/β-catenin signalling in cancer, which can contribute to tumour progression [28,35,36,37,38,39,40]. Here, we showed that *BCL9L* knockdown repressed only a subset of the analysed Wnt/β-catenin target genes, such as *AXIN2*, *LEF1*, and *BIRC5* in Cal29, especially after the activation of the signalling by the specific activator SKL2001. In T24 cells, *BCL9L* knockdown did not influence the analysed Wnt/β-catenin target genes. Given the observation that BCL9L has an oncogenic effect in both Cal29 and T24 cells, we hypothesised that the action of BCL9L might be dependent of Wnt/β-catenin signalling in Cal29 and independent of Wnt/β-catenin signalling in the case of T24 cells. However, it remains unclear how *BCL9L* promotes proliferation, migration and invasion of T24 cells.

Various BCL9L signalling pathways are suggested by other studies. *BCL9L* mostly affects a subset of Wnt/β-catenin target genes [28,31,38,50]. In contrast to this, other studies support that *BCL9**L* could mediate its function independent of Wnt/β-catenin signalling. For example, the mechanism of action of BCL9L protein appears to be dependent on molecular subtypes of breast cancer and cell lines. In oestrogen receptor alpha (ER) positive breast cancer, BCL9L mechanism is independent of Wnt/β-catenin, however, it is dependent on ER signalling [32].

In triple-negative breast cancer (TNBC), *BCL9L* mediates the malignancy of TNBC through Wnt/β-catenin and TGF-β signalling pathways [62]. The mentioned studies and our current study confirm that the mechanism of BCL9L protein might be dependent and independent of Wnt/β-catenin signalling.

Furthermore, we examined the functional role of Wnt/β-catenin signalling in BC, independent of BCL9L. It has been shown that β-catenin is more strongly expressed in high grade and invasive BC, and Wnt/β-catenin signalling is involved in the maintenance and regulation of urothelial cancer stem cells suggesting that β-catenin is associated with aggressiveness and poor prognosis [15,16,17]. β-catenin activation collaborates with other disturbed signalling pathways such as RAS activation and low expression of PTEN to enhance further the aggressiveness of BC [63,64]. In vitro studies have shown that knockdown of β-catenin inhibits the tumourigenicity, suggesting that β-catenin promotes bladder cancer [18,65]. We used the specific inhibitor iCRT3, which binds to β-catenin and interrupts the interaction to TCF/LEF transcription factors to inhibit the nuclear signalling from mimicking the effect *of BCL9L* knockdown [52]. iCRT3 treatment inhibited the tumourigenic properties of BC cells. This confirms that Wnt/β-catenin signalling has a tumourigenic effect on BC and further strengthens the hypothesis that BCL9L protein is able to induce malignancy of Cal29 cells through Wnt/β-catenin signalling because both BCL9L protein depletion and iCRT3 inhibit the malignant behaviour of Cal29 cells.

A reasonable question is why *BCL9L* influence on Wnt/β-catenin signalling is cell line specific in BC cells. This is addressed by the hypothesis that a certain Wnt/β-catenin threshold is required in order BCL9L to affect Wnt/β-catenin signalling. In hepatocellular carcinoma, it has already been shown that a certain Wnt/β-catenin signalling threshold is required to be functional [66]. Curtain cell lines might have a low-level Wnt/β-catenin signalling threshold, suggesting that the BCL9L/Wnt/β-catenin signalling axis is not completely functional. In line with this, our data showed that Cal29 is more responsive to the SKL2001 (Wnt/β-catenin activator) and iCRT3 (Wnt/β-catenin inhibitor) compared to T24 cells. This, in turn, could be translated as Wnt/β-catenin signalling is more functional in Cal29, and in the case of T24 this signalling, might not be properly functioning. This could explain why, for example, MMPs, especially MMP14 respond differently in Cal29 and T24 after iCRT3 treatment. Furthermore, the observation of differential localisation in BC cell lines could further support the involvement of BCL9L on Wnt/β-catenin signalling in Cal29 cells. Considering that Cal29 originates from a muscle-invasive metastatic bladder carcinoma and T24 from a non-muscle invasive and non-metastatic disease, it can be hypothesised that BCL9L/β-catenin signalling plays a crucial role later in the progression of BC. Lv et al. 2021 showed that BCL9L and β-catenin were highly expressed in a subpopulation of tumour cells that were very tumourigenic. They suggested the involvement of BCL9L protein and Wnt/β-catenin signalling in the self-renewal and pluripotent mechanisms of cancer stem cells [67]. Cancer stem cells are very resistant to chemotherapy, and it is because of these cells that the tumours can recurrent again after treatment [68,69]. In line with this, BCL9L/Wnt/β-catenin signalling might play a crucial role in the progression of BC through the maintenance of BC stem cells, suggesting that these BC stem cells be responsible for the heterogeneity of bladder cancer [17,70,71,72,73,74,75].

However, our results have to be confirmed by further experiments to analyse the function of BCL9L/β-catenin signalling in the progression of BC. For example, Yang et al. showed that in 3D spheroid model of T24 cells, β-catenin was significantly increased compared to 2D monolayer model [76]. Our next strategy is to examine in more detail the role of BCL9L/Wnt/β-catenin signalling in the invasiveness and progression of BC also EMT transition markers using 3D models such as spheroid and the ex vivo porcine bladder organ model as described elsewhere [77]. Nevertheless, we propose that BCL9L/β-catenin signalling might be an important pathway in the progression of BC. A therapeutic approach that targets cancer stem cells through BCL9L/Wnt/β-catenin pathway might be a promising strategy against bladder cancer.

## 4. Materials and Methods

### 4.1. Patients Material

Tumour tissue samples were obtained from patients subjected to transurethral resection of the bladder or radical cystectomy, and ureter samples were obtained from patients subjected to nephrectomy at the Department of Urology, Jena University Hospital, Germany. Samples for immunohistochemistry were formalin-fixed, embedded in paraffin for pathological examination, and stored at the Institute of Pathology, Jena University Hospital, Jena, Germany, until use. Frozen tissue from the same patients used for mRNA expression analysis originated from the biobank of the above-mentioned Department of Urology. The baseline characteristics (age, sex, tumour stage, and grade) of the bladder cancer subjects are summarised in Table 1. Healthy ureters originated from patients with polycystic kidney disease, renal angiomyolipoma and renal cell carcinoma, respectively. All patients gave their written informed consent to provide residual tissue for research. The biobank and the presented study were approved by the institutional ethical committee of Jena University Hospital (No. 3657-01/13; No. 4213-09/14). The study was conducted in accordance with the Declaration of Helsinki.

### 4.2. Immunohistochemical Analysis

Protein expression of tumour samples was analysed by immunohistochemical staining with Dako REAL EnVision Detection System K5007 (Agilent Technologies, Santa Clara, CA, USA) according to the manufacturer’s protocol against BCL9L (anti-BCL9L 1:100, HPA049370, Atlas Antibodies, Bromma, Sweden). Briefly, 4 µm sections from formalin-fixed paraffin-embedded (FFPE) tumour samples underwent deparaffinisation and the heat-induced epitope retrieval was done in Tris-EDTA Buffer (10 mM Tris Base, 1 mM EDTA, 0.05% Tween20, pH 9.0) at 98 °C for 20 min. A consecutive section was Hematoxylin–Eosin stained. Immunohistochemical staining was assessed by intensity: (no staining (−), weak (+), moderate (++) and strong (+++)) of the tumour cells. Moreover, the immunohistochemical staining was assessed by the H-Score. The H-Score is calculated by the formula: 3× percentage of strong staining + 2× percentage of moderate staining + percentage of weak staining [49]. The results were validated and confirmed by a uropathologist.

### 4.3. Cell Culture

The *human* bladder cancer cell lines T24 and TCCsup were cultured in RPMI 1640 medium with Glutamax (Thermo Scientific, Waltham, MA, USA) supplemented with 10% fetal bove serum (FBS), penicillin (100 U/mL) and streptomycin (100 µg/mL). The *human* bladder cancer cell line Cal29 was cultured in Dulbecco’s Modified Eagle Medium (DMEM) supplemented with 10% fetal bove serum (FBS) (Thermo Scientific, Waltham, MA, USA), penicillin (100 U/mL), and streptomycin (100 µg/mL). All cells were incubated in a humidified atmosphere at 37 °C with 5% CO_2_.

Short tandem repeat (STR) analysis was done by the Institute of Forensic Medicine, Jena University Hospital, Germany, using PowerPlex 16 System (Promega, Germany) for amplification and capillary gel electrophoresis for analysing allele profiles. Profiles were checked by the German Collection of Microorganisms and Cell Cultures online STR analysis.

### 4.4. UTR Mutation Analysis by Dual-Luciferase Reporter System Assay

For mutation analysis of untranslated regions (UTR), the luciferase reporter plasmid pmirNanoGLO (Promega, Fitchburg, WI, USA) was used. The whole wildtype and double mutated 5′ UTR sequences (A > T at position 743 and G > T at position 937, NM_182557.4) of *BCL9L* were synthesised by Eurofins Genomics Germany GmbH (Ebersberg, Germany) and cloned between EcoRV and BstEII restriction sites upstream of the *NLucP* luciferase gene in the pmirNanoGLO vector. The wildtype 3′ UTR fragment of 783 bp (1198-1986, NM_182557.4) was amplified using cDNA from HEK293 cells. Briefly, 2 μg of total RNA from HEK293 cells was reverse transcribed into cDNA using the GoScript Reverse Transcription System (Promega, Fitchburg, WI, USA) according to the manufacturer’s protocol, using oligo dT primers. The desired fragment of 783 bp was amplified by using specific cloning primers which contain the corresponding restriction sites (forward: gagaGCTAGCAAGTCGCTGCCAGGGCTG; reverse: gagaGTCGACGGCTCTGTGGGCTGGGT). The fragment was cloned between NheI and SalI restriction sites downstream of the *NLucP* generating the 3´ UTR wildtype plasmid. The fragment was also used as a template to generate the single mutated 3′ UTR of *BCL9L* (A > T at the position 1633, NM_182557.4) by using the Q5 site-directed mutagenesis´s protocol and specific mutagenesis primers (forward: CATTCTAGAC**T**GGGTGTCTTCTACCAG, reverse: GCCCTTGGTCCTGGAGGT). The mutated UTR was cloned in the pmirNanoGLO vector and the sequence was verified by Sanger sequencing.

For transfection, T24 or TCCsup cells were seeded in a 24-well plate and cultured until the cells reached about 80% confluency. Transfection with the wildtype and mutated UTR luciferase reporter plasmids using jetPRIME reagent was done according to the manufacturer’s protocol. Twenty-four hours post-transfection, the cells were lysated in 200 μL Passive Lysis Buffer (E1941, Promega, Germany) and 80 μL of the lysate was used for the measurement of the luciferase activity using the Dual-Glo Luciferase assay system (N1630, Promega, Fitchburg, WI, USA) on the Tecan Infinite M200 Pro reader (Tecan, Männedorf, Switzerland). Moreover, RNA was extracted in order to analyse the luciferase mRNA level. The NLucP luciferase activity and mRNA level was normalised to Luc2 luciferase to eliminate variations in the transfection efficiency (NLucP/Luc2). Fold change was calculated to emphasize the effects of mutation on luciferase activity and mRNA level. All experiments were repeated at least three times.

### 4.5. Total RNA Extraction

RNA extraction from tissues and transfected cells with siBCL9L or treated with iCRT3 or SKL2001 was performed as described elsewhere [41]. Briefly, micro-dissected tumour samples (patients: 1–8), which correspond to at least 75% tumour cell content or slices from three normal ureters were lysed in 1 mL Triazol reagent (Thermo Fisher Scientific, Waltham, MA, USA), followed by phase separation according to the manufacturer’s protocol. The aqueous phase, which contains RNA was thoroughly mixed with one volume of 70% ethanol, followed by RNA purification using a NucleoSpin RNA XS kit (Machery-Nagel, Düren, Germany) according to the manufacturer’s protocol. The total RNA from cell culture experiments was isolated using only Triazol reagent according to the manufacturer’s protocol, without any modification. The RNA was dissolved in nuclease-free water and stored at −80 °C.

The total RNA concentration was quantified using a Qubit 3.0 Fluorometer (Thermo Fisher Scientific, Waltham, MA, USA) according to the manufacturer’s protocols. The RNA quality was analysed by an Agilent 2200 TapeStation (Agilent Technologies, Santa Clara, CA, USA) instrument according to the manufacturer’s instruction.

### 4.6. cDNA Synthesis and Reverse Transcription Quantitative PCR (RT-qPCR)

2 μg of total RNA was transcribed into cDNA using the GoScript Reverse Transcription System kit (Promega, Fitchburg, WI, USA) according to the manufacturer’s protocol. The cDNA was diluted 1:8 with nuclease-free water and stored at −20 °C. The mRNA expression analysis was performed using a LightCycler 480 SYBR Green I master mix and the LightCycler 480 instrument (Roche Applied Science, Penzberg, Germany) utilising specific primers for each target (Appendix A). The Cp value was extracted using the second derivative maximum method from the LightCycler 480 software. The relative expression of the housekeeping genes *RPS13* and *RPS23* was calculated using the ΔΔCp method, including primer efficiency in the calculation [78,79].

### 4.7. Protein Analysis by Western Blotting

Western blotting was performed as described elsewhere [41]. Briefly, the cells were lysed in 5 volumes of cold NETN buffer (100 mM NaCl, 20 mM Tris/HCl pH 8.0, 1 mM EDTA, 0.5% NP-40) supplemented with protease inhibitors (10μg/mL Leupeptin and 1mM PMSF). 30 μg total protein was separated by 10% SDS-PAGE and blotted on a polyvinylidene fluoride (PVDF) using xCell II Blot Module (Thermo Fisher Scientific, Waltham, MA, USA). For immunological detection, the following antibodies were used: anti-BCL9L (HPA049370, Atlas Antibodies, Bromma, Sweden) and anti- α-tubulin (sc-5286, Santa Cruz Biotechnology, Dallas, TX, USA). Horseradish peroxidase-conjugated anti-*mouse* IgG-HRP (sc-516102, Santa Cruz Biotechnology, Dallas, TX, USA) and anti-*rabbit* IgG-HRP (sc-2370, Santa Cruz Biotechnology, Dallas, TX, USA) were used as secondary antibodies. The membrane was incubated with chemiluminescence (ECL) reagent (GE Healthcare, Chicago, IL, USA) and the produced light emission was detected by a GBox Chemi XX6 (Syngene, Cambridge, UK).

### 4.8. Protein Analysis by Immunofluorescence

For immunofluorescence, Cal29, T24 or TCCsup cells were seeded in 4-well Falcon culture slides (Corning Inc., Corning, NY, USA) and incubated in a humidified atmosphere at 37 °C with 5% CO_2_. After reaching 60–70% confluence, the cells were washed one time with PBS and incubated in pre-chilled (−20 °C) 100% methanol for 5 min at room temperature, followed by a blocking step in 1% BSA/10% normal *goat* serum/0.3 M glycine in 0.1% PBS-Tween for 1 h. Then, the cells were incubated with 1 μg/mL anti-BCL9L antibody (HPA049370, Atlas Antibodies, Bromma, Sweden) diluted in 1% BSA/0.1% PBS-T at 4 °C overnight. The AlexaFluor^®®^488 *goat* anti-*rabbit* antibody (ab150077, Abcam, Camebridge, UK) was used as a secondary antibody at 2 μg/mL diluted in 1% BSA/0.1% PBS-T for 1 h at room temperature in the dark, followed by DAPI (2.5 µM) incubation for 30 min in the dark. The slides were stored at 4 °C and were analysed by microscopy (40× magnification, AxioImager Z.2, Carl Zeiss AG, Oberkochen, Germany).

### 4.9. Wnt/β-Catenin Signalling Analysis

For the analysis of the effect of *BCL9L* on Wnt/β-catenin signalling, Cal29 and T24 cells were first transfected with 40 nΜ pooled siBCL9L (SI04198985, SI04178909, SI04135257 Qiagen, Hilden, Germany) or siControl (SI0365031, Qiagen, Hilden, Germany). 24 h after transfection, the cells were treated with 0, 20, and 40 μM SKL2001 (specific Wnt/β-catenin activator) in 0.1% DMSO, and after 2 days, the mRNA expression of Wnt/β-catenin target genes *AXIN2*, *LEF1*, *SP5*, *BIRC5*, *MMP9* and *MMP14* was analysed by qRT-PCR. For the inhibition of Wnt/β-catenin signalling, the Cal29 and T24 cells were treated with 0 µM, 20 μM and 40 μM iCRT3 (specific Wnt/β-catenin inhibitor) in 0.1% DMSO. After 2 days, the mRNA expression of Wnt/β-catenin target genes was analysed.

### 4.10. Proliferation and Spheroid Formation Assay

For proliferation assay, Cal29 or T24 cells were seeded on 6-well cell culture plates and after 24 h transfected with 40 nM pooled siBCL9L (SI04198985, SI04178909, SI04135257 Qiagen, Hilden, Germany) or negative control siRNA (SI0365031, Qiagen, Hilden, Germany) using INTERFERin reagent according to the manufacturer’s protocol (Polyplus-transfection, Illkirch, France). For inhibition of Wnt/β-catenin signalling, the cells were treated with 0, 20, or 40 μM iCRT3 in 0.1% DMSO. After 6 or 8 days, the cells were washed once in PBS buffer, fixed with 2% glutaraldehyde solution in PBS for 10 min and stained with 0.1% crystal violet solution for 30 min. Cells were gently washed with distilled water, dried overnight, and solubilised with lysing solution (0.1 M sodium citrate, 50% ethanol, pH 4.2) for 30 min. The absorbance was measured on an Infinite M200 Pro reader at 590 nm (Tecan, Männedorf, Switzerland).

For spheroid formation, T24 cells were gently trypsinated and collected at 100 g for 10 min. The cell pellet was resuspended with fresh growth medium, and 1000 cells were seeded in a cell repellent surface 96-well microplate (Greiner Bio-one, Frickenhausen, Germany). The plate was centrifuged at 100× *g* for 10 min and incubated in a humidified atmosphere at 37 °C with 5% CO_2_. After 24 h, the spheroids were treated with 0 µM, 20 μM or 40 μM iCRT3 in 0.1% DMSO. Images were taken using a Cell observer (Cellobserver Z1 with Colibri-7, Carl Zeiss, Oberkochen, Germany). The software MATLAB R20019a and AnaSP were used to extract the volume from spheroid formation experiments. All experiments for proliferation and spheroid formation were repeated at least three times.

### 4.11. Apoptosis Assay

Cal29 and T24 cells were transfected in 6-well plates, transfected with siRNA, or treated with iCRT3 as described above (proliferation and spheroid formation assay). Apoptosis was detected by an FITC Annexin V apoptosis detection kit (556547, BD Pharmingen, Franklin Lakes, NJ, USA) according to the manufacturer’s protocol. Data acquisition and analysis were performed by the BD Accuri C6 Plus software, using a specific template for the annexin V-FITC (FL1 channel, 530 nm filter) and PI (FL2 channel, 575 nm filter). At least 10,000 events were used for the analysis, excluding cell debris by setting appropriate light scatter gates. All experiments were repeated at least three times.

### 4.12. Migration and Invasion by xCELLigence Real-Time Cellular Analysis System

Migration and invasion of cells were analysed using the xCELLigence RTCA System (Agilent Technologies, Santa Clara, CA, USA) according to the manufacturer’s protocol (Cell Migration and Invasion protocol). The cells were transfected with 40 nM pooled siBCL9L or siControl on a 6-well plate for three days or treated with 40 µM iCRT3 in 0.1% DMSO for 2 days as previously described. The transfected or treated cells were trypsinated and resuspended in serum-free medium. For migration, 20,000 cells in 100 μL were added to the upper chamber of CIM -plates. In the case of invasion, 40,000 cells in 100 μL were added to the upper chamber wells, pre-coated with 1:12 or 1:40 diluted Matrigel (catalogue no. 354230, Corning, New York, NY, USA). In the lower chamber, medium with 10% serum was added. All experiments were repeated at least three times. For statistical analysis, the mean of three independent biologicals was combined and a two-tailed Student’s *t*-test was performed for the whole time frame.

### 4.13. Data and Statistical Analysis

Statistical analysis was performed using the software IBM SPSS Statistics Version 25 by combined independent biological replicates. Two-tailed unpaired Student’s *t*-test and the non-parametric Mann–Whitney *U*-test were performed to compare the two groups. The type of test that was used is mentioned specifically in the result part. A 95% confidence interval (*p*-value < 0.05) was considered as statistically significant (*).

## 5. Conclusions

BCL9L, a coactivator of β-catenin, was identified to be mutated at UTR in our patients with progressive BC. One mutation at 3′ UTR could influence the gene expression of *BCL9L*, providing an additional mechanism for the aberrant expression in cancer. This study showed for the first time that BCL9L is overexpressed in dysplastic urothelial cells and MIBC, suggesting an association with tumour stage and invasiveness. Moreover, in vitro analysis demonstrated that both Wnt/β-catenin signalling and its cofactor BCL9L promote malignant behaviour in BC cells, indicating an oncogenic role and involvement in BC progression. The oncogenic effect of *BCL9L* in BC cells could be dependent or independent of Wnt/β-catenin signalling. The development of specific inhibitors which interfere with the normal function BCL9L and Wnt/β-catenin might provide a potential therapeutic strategy against progressive BC.

## Figures and Tables

**Figure 1 ijms-23-05319-f001:**
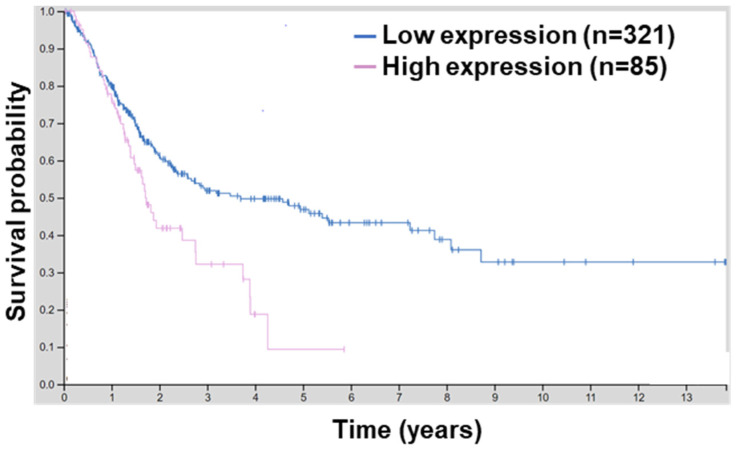
*BCL9L* expression is associated with worse survival probability. Survival analyses by the human protein atlas using TCGA data of 406 MIBC results in a significant correlation between high *BCL9L* expression and worse survival probability (expression cut-off 7.44 fragments per kilobase of transcript per million; log-rank *p* = 0.0029; http://www.proteinatlas.org/ENSG00000186174-BCL9L/pathology/urothelial+cancer) (accessed on 27 March 2022) [42,48].

**Figure 2 ijms-23-05319-f002:**
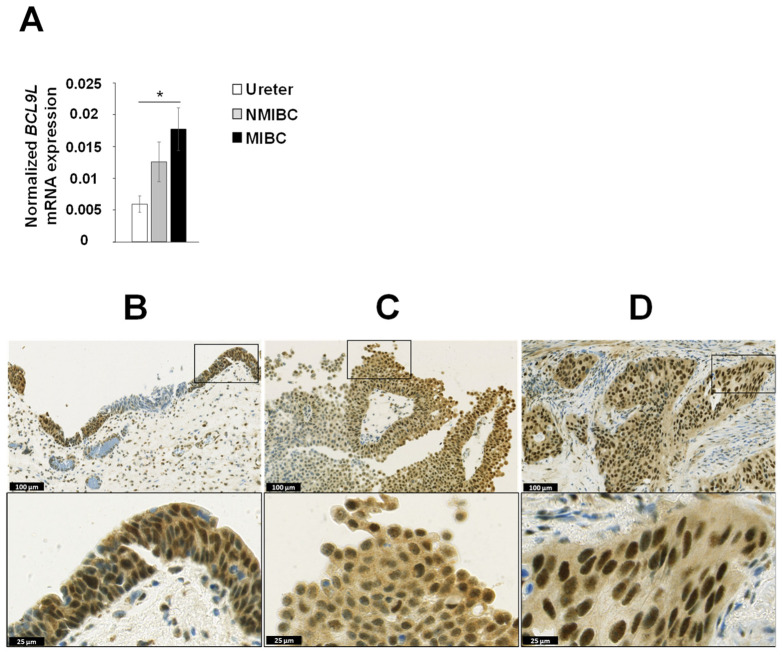
Expression of BCL9L in urothelium, dysplasia, and bladder cancer. (**A**) mRNA expression analysis of 8 matched NMIBC and corresponding MIBC patients and 3 normal urothelia (ureter) by qRT-PCR shows significantly increased expression in MIBC. The expression was normalised to housekeeping genes *RPS13* and *RPS23*. The data are expressed as mean ± standard deviation and statistical analysis was performed by non-parametric Mann–Whitney *U*-test with * *p*-value < 0.05. (**B**–**D**) Immunohistochemical staining of dysplastic urothelium, NMIBC, and MIBC. BCL9L is very heterogeneously expressed in different stages of urothelial dysplasia and bladder cancer. An increased expression was mainly found in dysplastic urothelium compared non or low dysplastic urothelium (B, patient 8), in peripheral cell layers of papillary NMIBC (C, patient 5), and in MIBC (D, patient 5). Scale bar upper images 100 µm, lower images 25 µm. Immunohistochemical staining of all patients are shown in Appendix A.

**Figure 3 ijms-23-05319-f003:**
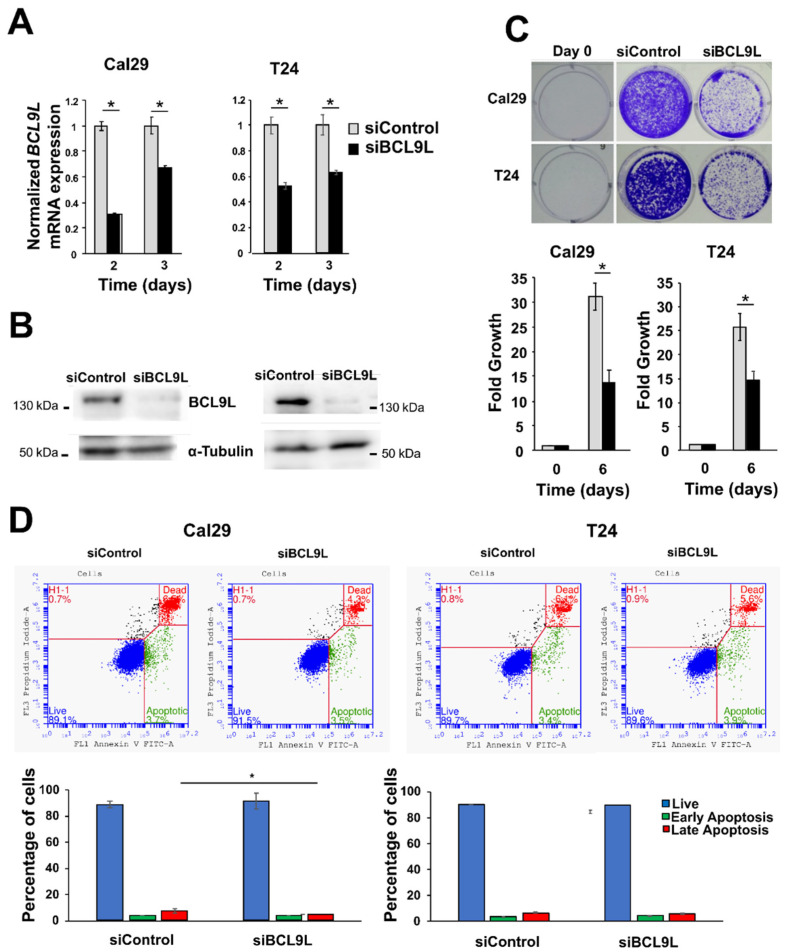
Cell proliferation and apoptosis analysis of bladder cancer after *BCL9L* knockdown. (**A**) mRNA expression of *BCL9L* after knockdown by qRT-PCR in Cal29 (*n* = 4) and T24 (*n* = 4) cells. siControl was set as 1 arbitrarily. (**B**) The protein level of BCL9L of transfected cells siBCL9L and siControl after 3 days transfection. (**C**) Cell proliferation is significantly reduced in BCL9L knockdown cells, analysed by crystal violet staining assay (*n* = 4) (**D**) *BCL9L* knockdown did not affect apoptosis of BC cells. The apoptosis assay was performed by dual staining with annexin V-FITC and propidium iodide (PI) kit and analysed by flow cytometry. * *p*-value < 0.05 Mann–Whitney *U*-test, *n* = independent experiments.

**Figure 4 ijms-23-05319-f004:**
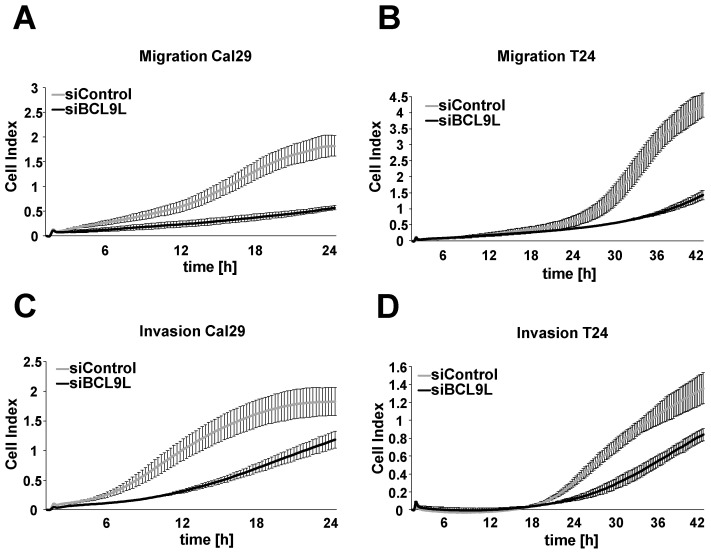
Knockdown of *BCL9L* represses migration and invasion of Cal29 and T24 cells. Transfected Cal29 (**A**) and T24 (**B**) cells with pooled siBCL9L or siControl were seeded on 16-well CIM plates, and the migration event was analysed in real-time by the xCELLigence system. Transfected Cal29 (**C**) and T24 (**D**) cells were seeded in the pre-coated with matrigel CIM plate and the invasion was analysed. The figure represents one of the biological replicates. All three independent biological replicates are shown in Appendix A. The data are expressed as mean ± standard deviation of two technical replicates. The cell index values correspond to the cell number that migrates or invades.

**Figure 5 ijms-23-05319-f005:**
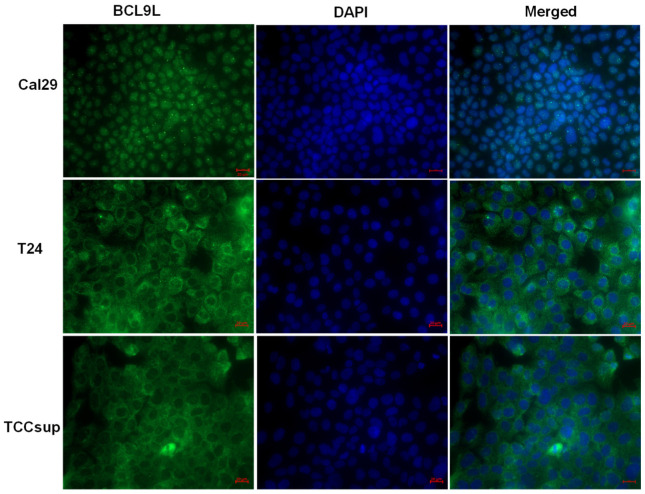
Intracellular localisation of BCL9L protein in bladder cancer cells. Immunofluorescence staining against BCL9L demonstrates that BCL9L is mainly localised in the nucleus of Cal29 and mainly in the cytoplasm of T24 and TCCsup cells. BCL9L is shown in green, detected by AlexaFluor^®®^488 *goat* anti-*rabbit* secondary antibody. Nuclear DNA was stained in blue with DAPI. In red is bar scale: 20 μm.

**Figure 6 ijms-23-05319-f006:**
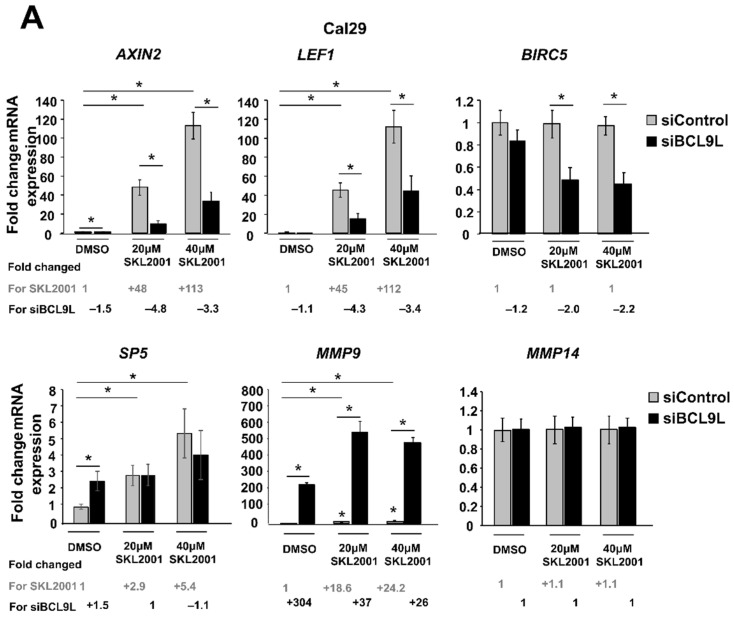
Repression of Wnt/β-catenin signalling by *BCL9L* knockdown is cell-line-specific in BC. (**A**,**B**) mRNA expression of Wnt/β-catenin target genes after siBCL9L and SKL2001 co-treatment Cal29 (*n* = 3) (**A**) and T24 (*n* = 3) (**B**). Cells were transfected with siBCL9L or siControl and next day were treated with Wnt/β-catenin activator (SKL2001) at different concentrations: 40 μΜ and 20 μΜ. 2 days later the mRNA expression of Wnt/β-catenin target genes *AXIN2*, *LEF1*, *SP5*, *BIRC5*, *MMP9*, and *MMP14* were analysed by qRT-PCR. Fold change and significance for different concentrations of SKL2001 compared to the negative control (DMSO + siControl) is indicated in grey. Fold change and significance of siBCL9L compared to each SKL2001 treatment or DMSO is indicated in black. The data are shown as relative to the negative control (DMSO + siControl) and are expressed as mean ± standard deviation. (* *p*-value < 0.05, Mann–Whitney *U*-test, *n*: independent biological replicates.

**Figure 7 ijms-23-05319-f007:**
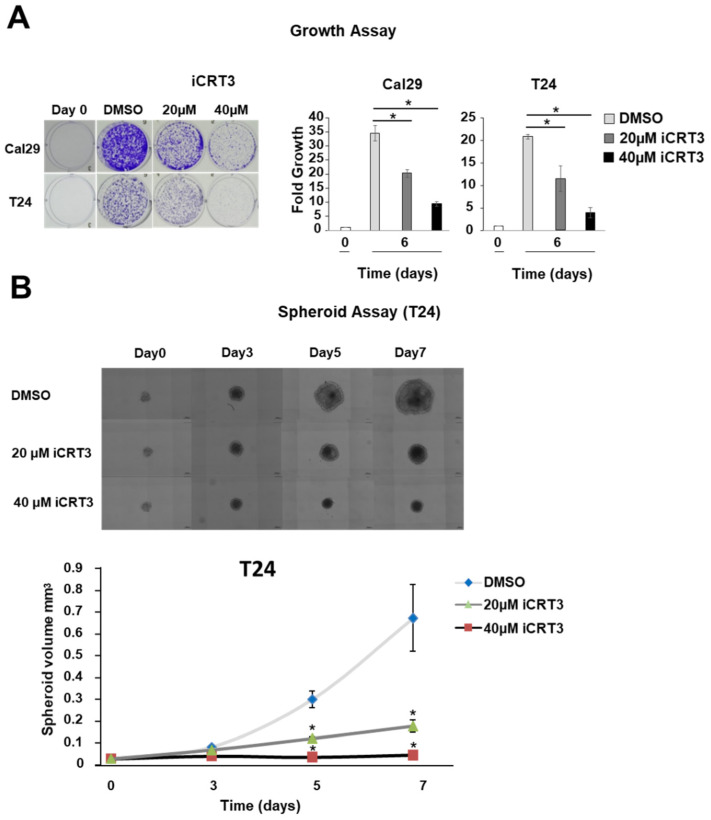
Proliferation and spheroid growth analysis after Wnt/β-catenin signalling inhibition in bladder cancer cells. (**A**) Crystal violet staining of Cal29 (*n* = 3) and T24 (*n* = 3) cells after 8 days treatment with 40 μM and 20 μM of the inhibitor iCRT3. Day 0 was set as 1 arbitrarily. (**B**) Spheroid analysis of T24 (*n* = 6) cells after 3-, 5-, 7-days treatment with 40 μM and 20 μM iCRT3. (* *p*-value < 0.05, Mann–Whitney *U*-test, *n*: independent biological replicates).

**Figure 8 ijms-23-05319-f008:**
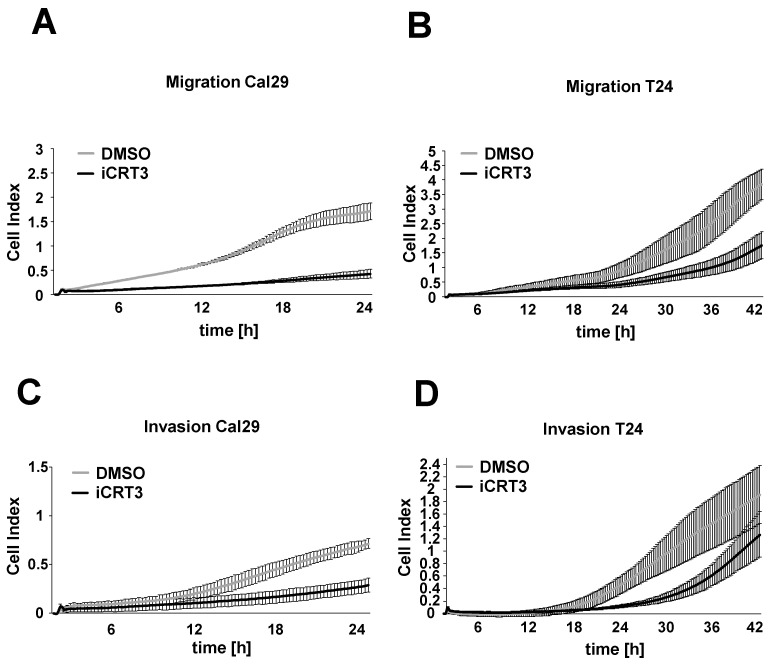
Cell migration and invasion analysis after Wnt/β-catenin signalling inhibition in bladder cancer cells. Treated Cal29 (**A**) and T24 (**B**) cells with 40 μM iCRT3 and 0.1% DMSO were seeded on CIM plates and migration was analysed in real-time by the xCELLigence system. Treated Cal29 (**C**) and T24 (**D**) cells were seeded in the pre-coated CIM plate with matrigel and the invasion was analysed. The figure represents one of the biological replicates. All three independent biological replicates are shown in Appendix A. The data are expressed as mean ± standard deviation of three technical replicates. The cell index values correspond to the cell number that migrates or invades.

**Table 1 ijms-23-05319-t001:** Patient material and characteristics. NMIBC, non-muscle-invasive bladder cancer; MIBC, muscle invasive bladder cancer; LG, low grade; HG, high grade; m, male; f, female.

Patient			Primary NMIBC	MIBC
No.	Age	Sex	TNM-T/Grading	TNM-T/Grading
1	67	m	pTa HG	pT4a HG
2	79	f	pT1 HG	pT2b HG
3	69	m	pTa HG	pT4a HG
4	78	f	pT1 HG	pT4a HG
5	65	m	pTa HG	pT2a HG
6	71	m	pT1 HG	pT2a HG
7	74	m	pTa HG	pT3a HG
8	73	m	pTa LG	pT2 HG
9	63	m	pTa HG	pT2a HG
10	76	m	pTa HG	pT3b HG
11	76	m	pTa HG	pT4a HG

## Data Availability

Data is contained within the article or Appendix A. The whole scans of IHC slides are available on request from the corresponding author.

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
