# Peer review of "Wnt/β-Catenin Signalling and Its Cofactor BCL9L Have an Oncogenic Effect in Bladder Cancer Cells"

_ijms, 2022, doi:10.3390/ijms23105319_

Round 1
Reviewer 1 Report
Dear Authors,
The paper is well written, I don't have many comments. The discussion is very lengthy I will really appreciate if you could reduce it little bit.
Here are some more comments.
1. In figure 1, the survival graph include a legend. Blue/Pink line represents what ?
2. In figure 2B, the scale is half missing and in Figure a B,C &D the scale in the IHC is not visible (too small)
3. In figure 3B, the tubulin band is very dull. Please repeat that one or put a new picture there.
4. When you claim knockdown of BCL9L reduces the migration and invasion of cancer cells, you need to show more data to support. Can you show the MMP-2 and 9 activity in gelatin Zymography? the reason I ask is that MMPs are secreted proteins and thay are activated later mRNA expression will not translate to its activity.
5. Can you show some evidence of EMT transition ? thay are prerequisites for invasion and migration.
6. Scale is missing in Figure 5, The image quality is very poor for a confocal microscope. AxioImages is not a confocal microscope. Please check the instrument.
Author Response
Dear Reviewer #1,
Thank you for reviewing our manuscript, your comments and constructive criticism. We hope that we have been able to answer all of your comments satisfactorily.
> The discussion is very lengthy I will really appreciate if you could reduce it little bit.
Thank you very much for the suggestion. We have reduced the length of the discussion in the revised manuscript.
> In figure 1, the survival graph include a legend. Blue/Pink line represents what ?
We have included the legend for blue/pink line in the graph.
> In figure 2B, the scale is half missing and in Figure a B,C &D the scale in the IHC is not visible (too small)
The scale bar in figure 2B was cut by mistake. We now included the whole image with scale bar. The scale bar for the Figure 2B, C and D is 100 μm (upper images). Now, we improved the visibility in the images of the revised manuscript.
> In figure 3B, the tubulin band is very dull. Please repeat that one or put a new picture there.
We have included new picture with higher exposure time and more intensity for the tubulin in case of Cal29 cells.
> When you claim knockdown of BCL9L reduces the migration and invasion of cancer cells, you need to show more data to support. Can you show the MMP-2 and 9 activity in gelatin Zymography? the reason I ask is that MMPs are secreted proteins and thay are activated later mRNA expression will not translate to ist activity.
Thank you very much for your suggestion. We are planning to analyse the role of BCL9L and Wnt/β-catenin signalling in a spheroid model and in a ex vivo porcine model system. For this reason, we want to generate stable knockdown (by lentivirus technology) or knockout for BCL9L (by CRISP/cas technology) cell lines and analyse in more detail the BCL9L/Wnt/β-catenin signalling, including protein expression of factors such as MMP2 and MMP9 by Immunostaining and we want to include in this setup up also gelatine zymography analysis.
> Can you show some evidence of EMT transition ? thay are prerequisites for invasion and migration.
In this project, we have not analysed EMT marker after siRNA transfection. We think that is more relevant to analyse EMT transition in 3D spheroid and ex vivo porcine model when we generate stable knockdown or knockout against BCL9L cell lines.
> Scale is missing in Figure 5, The image quality is very poor for a confocal microscope. AxioImages is not a confocal microscope. Please check the instrument.
We have included the scale bar in the image in the revised manuscript. Thank you for pointing the poor quality image problem. The designation “confocal microscopy” was wrong. The AxioImager Z.2 is not a confocal microscope. We deleted “confocal” in the methods part (line 706). We think that the images are sufficient to estimate BCL9L localisation in the cytoplasm or nucleus.
Best regards
Daniel Steinbach
Reviewer 2 Report
Kotolloshi et al., findings in this manuscript - suggested the oncogenic role of Wnt/β-catenin signaling and its cofactor BCL9L by investigating proliferation, migration and invasion of Bladder cancer cells.
Major/Minor comments to further improve the manuscript:
Page 1, Abstract: Authors mentioned that “Spheroid growth and proliferation were examined after knockdown by siBCL9L and the inhibition of Wnt/β-catenin signalling.”
- Spheroid growth investigation with knockdown of siBCL9L is missing in the manuscript.
Page 3, Figure 1:
- Please include a label that explains what blue and pink curves actually represent? and How many patients are included per group? – for example: High BCL9L expression (n=?) and Low BCL9L expression (n=??)
Page 3, lines 132-134: “The mRNA expression of BCL9L is significantly upregulated in MIBC compared to ureters from non-bladder cancer patients (2.9-fold induction, p<0.05, Figure 1B). ><0.05, (Figure 1B)”
- There is no Figure 1B in the manuscript – Change this to Figure 2A.
Page 4, line 140: “The BCL9L staining is observed both in the nucleus and in the cyto- 140 plasm but strongly intensified in the nucleus, especially in case of high expression”
- IHC images with insets to zoom in the nucleus staining might be useful to support the above statement!
Page 5, line 192: “The reduction of cell proliferation after BCL9L knockdown could be through induction of cellular apoptosis”.
- Decrease in cell proliferation could also be due to cell-cycle arrest (other possible mechanism). As authors didn’t observe any change in the apoptosis after siBCL9L or iCRT3 treatment, the decrease in the proliferation might be due to the defect in the cell-cycle after siBCL9L or iCRT3 treatment.
Page 7, lines 240-242 and Page 8, Figure 5: “BCL9L is mainly localised in the nucleus of Cal29 cells, however, T24 cells 240 shows more cytoplasmic localisation (Figure 5), suggesting a distinct role of BCL9L in 241 these cell lines.”
- A third cell line to show the localization of BCL9L might be very useful (which is more common - Cytoplasm Vs Nucleus localization of BCL9L)
- Please include a marker like F-Actin to show cytoplasm positive Staining
Page 9, Figure 6 and Supplementary Figure S5:
- SKL2001 and iCRT3 treatment showed inconsistent results related to MMP9 and MMP14 expression
- One would expect opposite expressions of MMPs with respect to these treatments. For example: T24 cells upon treatment with SKL2001 (Wnt/β-catenin Activator) and iCRT3 (Wnt/β-catenin Inhibitor) – showed increased MMP14 expression. This needs to be explained or discussed in the manuscript.
- Why Cal29 and T24 cells showed inconsistent results again with MMPs expression? In Supplementary Figure S5 - iCRT3 treatment increased MMP14 expression in T24 cells, whereas iCRT3 treatment decreased MMP14 expression in Cal29 cells. Is it related to the differential localization of BCL9L (Cytoplasm Vs Nucleus)?
Page 11, Figure 7: Spheroid growth assay
- Day 9 and Day 11 data – no need to include this, as the spheroids are big and cells within the spheroid are started dying in DMSO group.
- Label DMSO instead of Control (also in Supplementary Figure S5)
Pages 12-15, Discussion:
- Its too lengthy discussion. Authors explained/discussed each result again in the discussion. This is not necessary! Results need to be emphasized without going into much detail.
- EMT plays very crucial role in the migration/invasion of tumors – Which is missing from the manuscript discussion
- Also rescue experiments are missing from the study design to link BCL9L with Wnt/β-catenin signaling with respect to Bladder Cancer
Author Response
Dear Reviewer #2,
Thank you for reviewing our manuscript, your comments and constructive criticism. We hope that we have been able to answer all of your comments satisfactorily.
> Page 1, Abstract: Authors mentioned that “Spheroid growth and proliferation were examined after knockdown by siBCL9L and the inhibition of Wnt/β-catenin signalling.” Spheroid growth investigation with knockdown of siBCL9L is missing in the manuscript.
Thank you for this comment. The description in the abstract was misleading. We have rephrased the abstract to a general statement: “… cell proliferation was examined by crystal violet staining and by spheroid model” (line 14). The details are mentioned in the methods and results part.
> Page 3, Figure 1: Please include a label that explains what blue and pink curves actually represent? and How many patients are included per group? – for example: High BCL9L expression (n=?) and Low BCL9L expression (n=??)
Thank you for pointing out this mistake, we have included the legend of blue/pink line in the graph including number of cases. The number of cases with low expression (blue line) was n=321, and n=85 for high expression (red line).
> Page 3, lines 132-134: “The mRNA expression of BCL9L is significantly upregulated in MIBC compared to ureters from non-bladder cancer patients (2.9-fold induction, p<0.05, Figure 1B). ><0.05, (Figure 1B)” There is no Figure 1B in the manuscript – Change this to Figure 2A.
Thank you for pointing out this mistake, we have corrected it.
> Page 4, line 140: “The BCL9L staining is observed both in the nucleus and in the cyto- plasm but strongly intensified in the nucleus, especially in case of high expression” IHC images with insets to zoom in the nucleus staining might be useful to support the above statement!
We included a 4-fold zoom image below of the already shown image and marked the zoomed section, respectively.
> Page 5, line 192: “The reduction of cell proliferation after BCL9L knockdown could be through induction of cellular apoptosis”. Decrease in cell proliferation could also be due to cell-cycle arrest (other possible mechanism). As authors didn’t observe any change in the apoptosis after siBCL9L or iCRT3 treatment, the decrease in the proliferation might be due to the defect in the cell-cycle after siBCL9L or iCRT3 treatment.
We totally agree with this statement that the decrease of proliferation by siBCL9L and iCRT3 treatments could be because of growth arrest. By apoptosis we wanted to confirm if the cells are alive before performing migration and invasion experiments. We included the statement that “[…] maybe through cell-cycle arrest mechanism […]” in the discussion part, line 491.
> Page 7, lines 240-242 and Page 8, Figure 5: “BCL9L is mainly localised in the nucleus of Cal29 cells, however, T24 cells 240 shows more cytoplasmic localisation (Figure 5), suggesting a distinct role of BCL9L in 241 these cell lines.” A third cell line to show the localization of BCL9L might be very useful (which is more common - Cytoplasm Vs Nucleus localization of BCL9L) Please include a marker like F-Actin to show cytoplasm positive Staining
In the past we had included TCCsup in our experiments. We don´t have so many data with this cell line and we focused on Cal29 and T24. Nevertheless, BCL9L expression was also analysed by immunofluorescence in TCCsup. Here, BCL9L was mainly localized in the cytoplasm. We have included TCCsup as a third cell line in the revised manuscript. Unfortunately, we did not included F-actin in our experimental setup. Actually we don´t know at which cellular compartment (nucleus or cytoplasm) is more common BCL9L to be localised in bladder cancer cell lines. BCL9L immunostaining of BC tissues seems to be in both compartment localised, however it is stronger in the nucleus of tumour cells. We should keep in mind that localisation of BCL9L is analysed in 2D monolayer system of bladder cancer cells. We suspect that BCl9L/Wnt/β-catenin is more active in 3D model. In general we are planning to analyse in more detail the BCl9L/Wnt/β-catenin in 3D models such as spheroids and ex vivo porcine model system after stable or knockout BCL9L. We want also to investigate the localisation of BCL9L in such 3D model.
> Page 9, Figure 6 and Supplementary Figure S5: SKL2001 and iCRT3 treatment showed inconsistent results related to MMP9 and MMP14 expression. One would expect opposite expressions of MMPs with respect to these treatments. For example: T24 cells upon treatment with SKL2001 (Wnt/β-catenin Activator) and iCRT3 (Wnt/β-catenin Inhibitor) – showed increased MMP14 expression. This needs to be explained or discussed in the manuscript. Why Cal29 and T24 cells showed inconsistent results again with MMPs expression? In Supplementary Figure S5 - iCRT3 treatment increased MMP14 expression in T24 cells, whereas iCRT3 treatment decreased MMP14 expression in Cal29 cells. Is it related to the differential localization of BCL9L (Cytoplasm Vs Nucleus)?
Thank you for addressing this point. This is addressed by the Wnt/β-catenin signalling functionality in this cell lines. Our data showed clearly that Cal29 responses far stronger to the activator SKL2001 and inhibitor iCRT3 compared to T24. For us this is strong indication that the Wnt/β-catenin signalling is more functional or responsive in Cal29 cells compared to T24. In turn this might mean that Wnt/β-catenin signalling is not function properly in T24, also this is in agreement with our data. This can explain why MMPs, especially MMP14 responses differently. We discussed this in the revised manuscript in lines 553-554.
The observation of the differential localisation of BCl9L in Cal29 (nuclear localisation) and T24 (cytoplasm) could be a further explanation of why BCl9L influences a subset of Wnt/β-catenin targets genes in only in Cal29 and not in T24 cells. Again this might be in the same direction that in T24 cells the Wnt/β-catenin threshold is so low that the effect of BCL9L on all target genes is not visible anymore. That’s also the reason that we want to transfer the analysis in 3D model because we suspect that in such model the Wnt/β-catenin threshold is increased.
Also Wnt/β-catenin signalling is very dynamic and complicated signalling pathway. Moreover, the Wnt/β-catenin target genes were defined in other cell line and cancer types and some of them are also regulated by other transcription factor independent of Wnt/β-catenin signalling. So we don´t know how they are regulated in bladder cancer. It is not surprisingly that some Wnt/β-catenin target genes could behave unexpectedly in just treatments. For example, knockdown of BCL9L reduced a subset of Wnt/β-catenin target genes in colon cancer, however, one target gene MSX2 is induced after BCL9L knockdown (Brembeck et al. 2011. Gastroenterology 141(4): 1359-1370.e1353) . Another example, SKL2001 treatment increased Axin2 mRNA level (expected), however reduced cMyc mRNA level (unexpectedly), which is another Wnt/β-catenin target gene (Ohashi et al. 2017. Biochem Biophys Res Commun 493(3): 1342-1348).
> Page 11, Figure 7: Spheroid growth assay Day 9 and Day 11 data – no need to include this, as the spheroids are big and cells within the spheroid are started dying in DMSO group. Label DMSO instead of Control (also in Supplementary Figure S5)
Thank you for your suggestion. We have excluded the day 9 and 11 in the revised manuscript. We also have labelled DMSO instead of control.
> Pages 12-15, Discussion: Its too lengthy discussion. Authors explained/discussed each result again in the discussion. This is not necessary! Results need to be emphasized without going into much detail.
We reduced the discussion part in the revised manuscript.
> EMT plays very crucial role in the migration/invasion of tumors – Which is missing from the manuscript discussion.
Thank you for pointing out this. We are planning to analyse in more detail the role of BCL9L/β-catenin in 3D model such as spheroids and ex vivo porcine bladder organ model after generation stable knockdown or knockdown cell lines. In this new experimental setup we want to analyse the EMT markers and other factors as we think it makes more sense to transfer the analysis of BCL9L and Wnt/β-catenin signalling in 3D model. We discussed this in the revised manuscript in lines 492-495.
> Also rescue experiments are missing from the study design to link BCL9L with Wnt/β-catenin signaling with respect to Bladder Cancer
Unfortunately, we had not included rescue experiments. It is actually a good suggestion to include such experiments when we transfer the analysis in 3D models with stable transfected bladder cells, described above.
Best regards
Daniel Steinbach
Round 2
Reviewer 2 Report
The authors have addressed my concerns adequately.